# Production of an Anise- and Woodruff-like Aroma by Monokaryotic Strains of *Pleurotus sapidus* Grown on *Citrus* Side Streams

**DOI:** 10.3390/molecules27030651

**Published:** 2022-01-19

**Authors:** Friederike Bürger, Maximilian Koch, Marco A. Fraatz, Alejandra B. Omarini, Ralf G. Berger, Holger Zorn

**Affiliations:** 1Institute of Food Chemistry and Food Biotechnology, Justus Liebig University Giessen, Heinrich-Buff-Ring 17, 35392 Giessen, Germany; friederike.buerger@lcb.chemie.uni-giessen.de (F.B.); maximilian.koch@lc.chemie.uni-giessen.de (M.K.); marco.fraatz@lcb.chemie.uni-giessen.de (M.A.F.); 2Fraunhofer Institute for Molecular Biology and Applied Ecology, Ohlebergsweg 12, 35392 Giessen, Germany; 3CONICET Asociación para el Desarrollo de Villa Elisa y Zona Héctor de Elia 1247, Villa Elisa E3265, Entre Ríos, Argentina; alejandra.omarini@adesarrollo.com.ar; 4Institute of Food Chemistry, Leibniz University Hannover, Callinstrasse 5, 30167 Hannover, Germany; rg.berger@lci.uni-hannover.de

**Keywords:** monokaryons, *Pleurotus sapidus*, aroma extract dilution analysis, *p*-anisaldehyde, (2*S*)-hydroxy-1-(4-methoxyphenyl)-1-propanone

## Abstract

The production of natural flavors by means of microorganisms is of great interest for the food and flavor industry, and by-products of the agro-industry are particularly suitable as substrates. In the present study, *Citrus* side streams were fermented using monokaryotic strains of the fungus *Pleurotus sapidus*. Some of the cultures exhibited a pleasant smell, reminiscent of woodruff and anise, as well as herbaceous notes. To evaluate the composition of the overall aroma, liquid/liquid extracts of submerged cultures of a selected monokaryon were prepared, and the volatiles were isolated via solvent-assisted flavor evaporation. Aroma extract dilution analyses revealed *p*-anisaldehyde (sweetish, anisic- and woodruff-like) with a flavor dilution factor of 2^18^ as a character impact compound. The coconut-like, herbaceous, and sweetish smelling acyloin identified as (2*S*)-hydroxy-1-(4-methoxyphenyl)-1-propanone also contributed to the overall aroma and was described as an aroma-active substance with an odor threshold in air of 0.2 ng L^−1^ to 2.4 ng L^−1^ for the first time. Supplementation of the culture medium with isotopically substituted l-tyrosine elucidated this phenolic amino acid as precursor of *p*-anisaldehyde as well as of (2*S*)-hydroxy-1-(4-methoxyphenyl)-1-propanone. Chiral analysis via HPLC revealed an enantiomeric excess of 97% for the isolated product produced by *P. sapidus.*

## 1. Introduction

With about 200 million tons in 2019, *Citrus* fruits belong to the most important plant species cultivated worldwide [1]. After the production of *Citrus* juice and extraction of essential oils, the peels remain as a by-product. Their current utilization for pectin extraction or animal feed is economically only modestly attractive. Nevertheless, these by-products contain numerous bioactive compounds [2], which may be transformed through fermentation and thus channeled back into food production. An alluring option is the production of flavorings, whereby the biotechnological route leads to natural flavors, which offers a considerable economic advantage.

Several studies, in which industrial or agricultural by-products, such as *Eucalyptus* waste, molasses, and apple pomace were used for the production of natural flavors have been reported [3,4,5]. Fungi from the division basidiomycota have become a focus of research, as they have a highly complex enzyme system, which enables them to degrade difficult-to-access lignin, cellulose, and hemicellulose-containing substrates. According to their preference for cellulose or lignin degradation, respectively, brown-rot and white-rot fungi are distinguished. Involved in lignin degradation are mainly laccases (EC 1.10.3.2) and various peroxidases (EC 1.11.1.X). An intensely studied representative of white-rot fungi is *Pleurotus sapidus* (PSA), in which a large number of enzymes of the ligninolytic system have been identified [6,7,8,9,10].

Only recently, a dye-decolorizing peroxidase (DyP) of PSA was found to cleave alkenes efficiently, and monokaryotic strains (Mk) partly showed improved enzyme activities when compared to the corresponding dikaryotic strain (Dk) [11]. Economically important flavoring substances, such as vanillin, *p*-anisaldehyde, and 3,4-dimethoxybenzaldehyde may be produced via lignin degradation [12].

Apart from lignocelluloses, a large number of compounds with terpenoid structures are present in *Citrus* peels, which represent potential precursors for biotransformation. For example, the bioconversion of *β*-myrcene to the citrus-like, flowery-smelling perillene was achieved with a close relative of PSA, the oyster mushroom *Pleurotus ostreatus* [13].

The aroma profile of the fermentation products is completed via the *de novo* synthesis of the fungi. PSA is able to synthesize, among others, 3,6-dimethyl-2,3,3a,4,5,7a-hexahydrobenzofuran (dill ether) and 3,6-dimethyl-3a,4,5,7a-tetrahydro-1-benzofuran-2(3*H*)-one (wine lactone), but also *p*-anisaldehyde [4,12,14,15]. According to studies with other basidiomycota, the formation of the latter substance and related structures starts from the aromatic amino acids l-phenylalanine and l-tyrosine, whereby different formation pathways have been postulated [16,17,18].

The aim of the current study was to make use of the great potential of precursors of *Citrus* side streams for fermentative aroma production using di- and monokaryotic strains of PSA, to identify relevant odor-active compounds and to gather more information regarding their formation pathways. Of particular importance was the acyloin, identified as (2*S*)-hydroxy-1-(4-methoxyphenyl)-1-propanone (2-HPP), which, to the best of our knowledge, has not been described as an odor-active substance before.

## 2. Results and Discussion

### 2.1. Screening of Di- and Monokaryotic Strains of Pleurotus sapidus

In order to investigate differences in aroma formation between di- and monokaryons of PSA, the strains were grown in surface cultures on an agar medium containing *Citrus* side stream substrate (CSSM). When the media were completely covered by the mycelium, an intense anise, marzipan, and vanilla-like aroma was noticeable. These odor impressions were similar to those of PSA cultures grown submerged in isomaltulose molasses medium [4]. In addition, some of the investigated strains were also described with odor attributes such as woodruff-like, tonka bean, and herbaceous. The odor intensities differed between the various Mk. Some cultures were also described as unpleasant and musty. Accordingly, the strains were classified into four aroma categories (Appendix A). Since submerged cultures are easier to handle on an industrial scale than surface cultures, the system was transferred from agar plates to shaking flasks. The sensory analysis of liquid-liquid extracts (LLE) with subsequent solvent-assisted flavor evaporation (SAFE) of selected interesting strains gave a more defined picture of the differences between the strains (Figure 1). The intensity for the descriptor “citrus” was in all cultures ≤ 0.5 and is therefore not shown.

The most intense notes reminiscent of woodruff, aniseed, butter, and vanilla were elicited by the dikaryotic strain, but the flavor extract obtained from this culture was hardly perceived as fruity and herbaceous. Mk 74 was described as woody, in addition to the aniseed and vanilla notes. This attribute, in a toned-down form, was otherwise only present for Mk 93. An especially overall appealing aroma profile was perceived from cultures of Mk 37, and this strain was thus selected for the further studies.

To the best of our knowledge, only very few studies have been published on comparative aroma analyses of di- and monokaryons [19]. Recently published results showed that different monokaryotic strains of PSA were able to produce higher concentrations of *p*-anisaldehyde via cleavage of *trans*-anethole than the dikaryotic strain. The authors showed that the concentration of *p*-anisaldehyde correlated directly with the peroxidase activity, which means that there were some monokaryons with higher enzyme activity than the parental strain [11].

### 2.2. Aroma Extract Dilution Analysis and Compound Identification

An aroma extract dilution analysis (AEDA) of the SAFE distillates was carried out for Mk 37 to identify key aroma compounds. By means of AEDA, 48 odor impressions with FD ≥ 64 were perceived at the olfactory detection port (ODP) (Table 1, Figure 2). Predominant aroma notes were “herbaceous”, “sweetish”, “citrus”, “terpenic”, and “flowery”.

A number of monoterpenes and terpene alcohols, which typically occur in *Citrus* fruits [20,21,22,23,24] were detected. The isomers *α*-pinene and *β*-pinene (**1**, FD 128; **4**, FD 512) were found besides sabinene (**5**, FD 2048), (*R*)-limonene (**6**, FD 256), and *p*-cymene (**8**, FD 8192). Furthermore, terpinen-4-ol (**21**, FD 1024), the corresponding isomer *α*-terpineol (**25**, FD 16,384), as well as the “flowery” and “fruity” smelling compound linalool (**19**, FD 2048) were detected. All of these compounds were also identified in the *Citrus* medium used for the cultivation of PSA Mk 37. Limonene and linalool have previously been identified in submerged cultures of PSA [14]. Therefore, the monoterpenes’ origin might be attributed to both the culture substrate and the fungal metabolism.

Other aroma compounds reported for *Citrus* fruits, such as *p*-cymen-8-ol (**29**, FD 512) and the two monoterpene ketones dihydrocarvone (**22**, FD 1024) and piperitone (**26**, FD 64) [22,24], were not present in the used substrate and therefore originated from the fungal metabolism.

Compound **22** has been described as a conversion product of myrcene by various strains of basidiomycota, including PSA [25]. Since myrcene was detected in CSSM, it is most likely the precursor of dihydrocarvone, which has been described in literature as minty and fresh [25].

(*Z*)-Linalool oxide (furanoid) (**16**, FD 64) often occurs together with linalool and has been described frequently for white rot fungi [15,26,27]. In contrast, to our knowledge there are no reports on the occurrence of (*Z*)-(+)-limonene oxide (**15**, FD 1024) in basidiomycota. It has been described as a component of *Citrus* peel or oils but was not detected in the CSSM substrate [28,29,30].

A compound with a comparatively high FD-factor was the bicyclic monoterpenoid dill ether 1 (**17**, FD 32,768), while dill ether 2 showed a much lower FD factor (**20**, FD 128). The biosynthesis of these monoterpene ethers as well as of wine lactone (**39**, FD 2048) by PSA has been reported by Trapp et al., who have shown that these benzofuranoid compounds were formed de novo by the fungus from d-glucose [14]. The biosynthesis of dill ether has also been reported for the fungus *Cystostereum murrayi* recently [31].

Another important group of identified aroma active substances comprised arylic compounds including *p*-anisaldehyde (**33**, FD 262,144). This compound, as well as (*E*)-methyl cinnamate (**34**, FD 1024) and methyl *p*-anisate (**35**, FD 4096), is well known for cultures of PSA [4,12,15]. In the culture extract of CSSM PSA Mk 37, additional aromatic compounds, including anisole (**11**, FD 1024), benzaldehyde (**18**, FD 1024), *p*-methylacetophenone (**27**, FD 2048), methyl *p*-anisate *(***35***,* FD 4096)*, p*-methoxyphenylacetone (**37**, FD 512), *p*-methoxypropiophenone (**38**, FD 1024), *p*-methoxybenzyl alcohol (**40**, FD 512), 3,4-dimethoxybenzaldehyde (**43**, FD 8192), and 2-hydroxy-1-(4-methoxyphenyl)-1-propanone (**48**, FD 512), were detected (Figure 3).

The structural similarity of these compounds was reflected in similar olfactory impressions. Thus, a sweetish odor was perceived for all arylic compounds except for compound **37**, which has been described as “fruity” in the literature [32]. In contrast, this substance was perceived as green, woody, and spicy in the sensory evaluation. AEDA indicated that *p*-anisaldehyde (**33**, FD 262,144) was a key aroma compound, contributing to the distinct sweetish, anisic, and herbaceous notes of the overall aroma. In plants, *p*-anisaldehyde is found in fennel, aniseed, and also in vanilla [32,33,34,35,36,37,38,39]. Depending on the study, 0.3 mg kg^−1^ to 132 mg kg^−1^ of *p*-anisaldehyde have been quantified in vanilla beans [37,38,39]. Takahashi et al. have identified the key aroma compounds of Tahitian cured vanilla beans and have determined a FD factor of 1,953,125 for both vanillin and *p*-anisaldehyde. Thus, both contribute significantly to the overall aroma of vanilla [38].

Compound **48** could not be identified via comparison to the National Institute of Standards and Technology (NIST) database. After isolation by means of column chromatography and preparative high performance liquid chromatography (HPLC), the structure was elucidated via gas chromatography-mass spectrometry (GC-MS) (Appendix A), high-resolution mass spectrometry (HRMS), and NMR spectroscopy as 2-hydroxy-1-(4-methoxyphenyl)-1-propanone.

NMR, GC−MS, and HRMS data of 2-HPP (**48**): ^1^H NMR (CDCl_3_, 400 MHz): *δ* 1.44 [3 H, d, *J* = 7.0 Hz, H-C(3)], 3.84 (1 H, d, *J* = 6.4 Hz, -OH), 3.89 (3 H, s, -OMe), 5.11 [1 H, q, *J* = 6.8 Hz, H-C(2)], 6.98 [1 H, d, *J* = 8.9 Hz, H-C(3′)], 7.93 [1 H, d, *J* = 9.0 Hz, H-C(2′)]. ^13^C NMR (CDCl_3_, 100 MHz): δ 200.7 C(8), 164.2 C(4′), 131.0 C(2′), 126.1 C(1′), 114.1 C(3′), 68.9 C(2), 55.6 C(OMe), 22.7 C(3). GC−MS (EI, 70 eV): 107 (6), 109 (4), 135 (100), 136 (11), 137 (19), 180 (3). HRMS (ESI−TOF): 203.0673 Da, −2.8 ppm deviation from the theoretical mass (203.0679 Da C_10_H_12_O_3_Na^+^). The NMR data (Appendix A) reported here are in agreement with those reported in the literature [40,41,42]. The deviations between the determined accurate masses and the theoretical accurate masses were <5 ppm.

Chiral analysis via HPLC revealed an enantiomeric excess of 97% for the isolated product produced via PSA (Appendix A). 2-HPP formed by baker’s yeast revealed an enantiomeric excess of 99% for the same enantiomer. The specific optical rotation of (−28.9 ± 0.8)° indicated the presence of the (*S*)-enantiomer via comparison with reported data [40].

2-HPP has been identified in cultures of a relative of PSA, *Pleurotus pulmonarius*, to show a comparatively high nematicidal activity towards *Caenorhabditis elegans* [43]. However, there are no previous reports on the aroma properties of 2-HPP available in the literature.

Compound **42** was the compound with the second highest FD factor of 65,536 (flowery, sweetish, herbaceous), but could not be instantly identified. However, based on the mass spectrum (Appendix A), it was concluded that it represents a benzofuranone derivative. Relevant fragments with the corresponding relative intensities were *m/z* 111 (100), 137 (43), and 180 (28). These are also characteristic of 5,6,7,7a-tetrahydro-4,4,7a-trimethyl-2(4*H*)-benzofuranone [*m/z* 111 (100), 137 (59), 180 (34)] (Appendix A), but the RI values on the polar column did not match. Wine lactone **39** also belongs to this class of compounds and is characterized by a low odor threshold (up to 0.01 pg L^−1^ in air) [44]. This may also be true for compound **42**, as the peak area was very small.

### 2.3. Stable Isotope Dilution Analysis

Based on the results of the AEDA, the concentration of the key aroma compound **33** was determined in cultures of CSSM PSA Mk 37 on three different days (4, 6, and 8). In a preliminary experiment, kinetics for the formation of flavoring substances in the CSSM were recorded (data not shown), which showed that the fungus formed high amounts of *p*-anisaldehyde from culture day 4 on, which decreased after day 8.

Evaluation of the stable isotope dilution analysis (SIDA) was done via the base peak ([M-H^+^], *m/z* 135) of *p*-anisaldehyde. The mass spectrum of the substituted compound showed a shift of one mass unit in this fragment ([M*-H^+^], *m/z* 136) (Figure 4).

The MS response factor was determined to be 0.89 according to Steinhaus et al. with the modification that mass ratios from 1:4 to 4:1 were used in the present study [45] (Appendix A). Depending on the culture day, *p*-anisaldehyde concentrations of 9.2 mg L^−1^ to 160.3 mg L^−1^ were determined (Table 2).

PSA Mk 37 formed the highest concentrations of *p*-anisaldehyde on culture day 8 with *Citrus* side stream as substrate.

As early as 1994, Abraham and Berger determined the concentration of *p*-anisaldehyde in a culture with PSA at 13 mg L^−1^ in a chemically defined medium [15]. In cultures of *P. pulmonarius*, about 27 mg L^−1^ *p*-anisaldehyde were determined in a glucose-based medium (on culture day 12) [12]. A study by Gutiérrez et al. showed that all *Pleurotus* spp. examined except for *P. eryngii* formed *p*-anisaldehyde in a lignin-based medium [12]. Using the lignin model substance dihydroanisoin, Shimada and Gold reported that the fungus *Phanerochaete chrysosporium* cleaves the diol moiety to produce anisaldehyde as an initial product, which further reacts to the corresponding alcohol (**40**), which was also detected in PSA [46].

### 2.4. Supplementation with l-Tyrosine and Formation of 2-HPP

In a study from 2002, different concentrations of l-tyrosine (1–10 mM) were added to cultures of *P. ostreatus* (oyster mushroom). With 5 mM added, an anisaldehyde concentration of 366 mg L^−1^ was reached 18 days after inoculation in static culture. A further increase in tyrosine concentration to 10 mM did not lead to higher levels within the investigated time period [47]. Since PSA is a close relative of *P. ostreatus*, it was interesting to investigate whether supplementation with l-tyrosine would also trigger the formation of other aromatic flavor compounds. The chromatograms for culture day 4 after supplementation showed a very intensive peak for *p*-anisaldehyde. The concentration as determined via SIDA was (25.6 ± 4.3) mg L^−1^ and increased to (71.1 ± 24.2) mg L^−1^ on day 8. Okamoto et al. compared static cultivation with shaking cultivation, where the anisaldehyde level remained the same in the latter over the culture period [47]. This observation could not be confirmed by our studies on PSA, but it might be possible to increase the concentrations of this key aroma substance by applying a static system as well.

To support the hypothesis that l-tyrosine is involved in the biosynthesis of *p*-anisaldehyde in PSA, a supplementation experiment with deuterated tyrosine was carried out. As expected, the addition of l-2-amino-3-([3,5-^2^H]-4-hydroxyphenyl)-propanoic acid resulted in a mass shift of +2 for anisaldehyde (*m/z* 138 for [3,5-^2^H]-4-methoxybenzaldehyde; Appendix A), while the non-labeled aldehyde showed a molecular ion of *m/z* 136.

Moreover, substituted 2-HPP was identified in the GC chromatogram of the culture extract after the addition of deuterated l-tyrosine. The formed 2-HPP without substitution showed a molecular ion of *m/z* 180 with a relative intensity of 3% and *m/z* 181 of <1%. After supplementation with l-2-amino-3-([3,5-^2^H]-4-hydroxyphenyl)-propanoic acid, the relative intensity of *m/z* 180 was 4%, *m/z* 181 was 1%, *m/z* 182 was 2%, and *m/z* 183 was <1% (Appendix A).

The results thus clearly indicated l-tyrosine as precursor for both *p*-anisaldehyde and 2-HPP. In the literature, the formation pathway of this acyloin in basidiomycota has not fully been elucidated. In the well-studied white rot fungus *Bjerkandera adusta*, compound **48** was detected after supplementation of the culture medium with *p*-anisaldehyde (ring ^13^C_6_-substituted) using gas chromatography-mass spectrometry [48]. However, the main focus of these investigations was on chloroarylpropane diols, some of which had already been described in 1998 by Swarts et al. for species of *Bjerkandera* [49]. Silk et al. also detected such diols after supplementation with ^13^C_9_-l-phenylalanine [50]. In the CSSM PSA Mk 37 extract 1-(4-methoxyphenyl)-1,2-propanediol was tentatively identified and the corresponding diketone (1-(4-methoxyphenyl)-1,2-propanedione) was identified, but both compounds did not show FD values > 64. The latter was not identified in the *Bjerkandera* studies. The biosynthesis of the diol is attributed to a reductive pathway in which the acyloin is regarded as an intermediate [50,51]. On the other hand, there is evidence that the diol is oxidized to the α-hydroxyketone during lignin degradation [52]. Additionally, aryl diketones have already been isolated as degradation products of lignin, so that these structures served as lignin model substances [53,54]. Following this approach, 1-(4-methoxyphenyl)-1,[2-^13^C]-propanedione (Appendix A) was transformed with PSA analogous to the reduction of α-diketones using Baker’s yeast [55]. This resulted in the detection of the substituted 2-HPP with a molecular ion of *m/z* 181 (Appendix A) and underlines the hypothesis that the formation of 2-HPP is related to the ligninolytic system of the fungus. Aldo-keto reductases, which to the best of our knowledge have not yet been described in PSA, may be responsible for this reduction in other fungi of the division basidiomycota [56]. This still offers considerable research potential on the molecular level.

### 2.5. Odor Threshold of 2-HPP

Since there are only a few studies available on the odor qualities of acyloins and the odor of 2-HPP has not been described so far, the odor threshold was determined using gas chromatography-flame ionization detection-olfactometry (GC-FID-O). Compound **48** exhibited a coconut-like, herbaceous, and sweetish odor with a threshold in air of 0.2 ng L^−1^ to 2.4 ng L^−1^. Neuser et al. investigated odor-active α-hydroxy ketones, but most of them were aliphatic. The only aromatic compound evaluated in this study, 3-hydroxy-4-phenyl-2-butanone (floral, sweet), had an odor threshold of 75 to 100 ng [57]. 2-HPP was 80- to 1000-fold more potent (0.1 to 0.9 ng absolute). Other acyloins produced from terpenoids also had significantly higher odor thresholds (110 to 1000 ng) [58].

### 2.6. Proposed Pathways

Our work focused on the two aroma-active compounds *p*-anisaldehyde (**33**) and 2-HPP (**48**), which were formed during the cultivation of PSA on CSSM. Our studies revealed that l-tyrosine was an optional precursor for both metabolites in the *de novo* synthesis. This is in agreement with studies in *Bjerkandera*, where this has been shown for *p*-anisaldehyde [59,60]. More often, however, the described postulated biosynthetic pathways for *p*-anisaldehyde start from phenylalanine [16,17,18]. Here, according to Lapadatescu et al., *p*-hydroxybenzaldehyde is assumed to be the direct precursor [18]. The transfer of a methyl group by an *O*-methyltransferase is likely, but no investigations have been carried out for this substrate in white rot fungi so far. An enzyme that catalyzes the formation of α-hydroxyketones is pyruvate decarboxylase (PDC, EC 4.1.1.1). From biotransformation studies with PSA starting from pyruvate and *p*-anisaldehyde, mainly 1-hydroxy-1-(4-methoxyphenyl)-2-propanone (1-HPP, derivative of phenylacetylcarbinol) is formed instead of 2-HPP (unpublished results), which indicates that this enzyme plays a minor role in the biosynthetic pathway. Alternatively, a benzaldehyde lyase (BAL, 4.1.2.38) would also lead to the formation of acyloins, but so far only the (*R*)-enantiomers have been described as products of this enzyme [61]. More likely is the route via a benzoylformate decarboxylase (BFD, EC 4.1.1.7), in this case starting from (4-methoxyphenyl)glyoxylic acid. This enzyme plays an important role in the mandelate pathway and resulted in the (*S*)-enantiomer of 2-HPP [62]. Thus, in addition to the more frequently described phenylalanine ammonia lyase (PAL)/tyrosine ammonia lyase (TAL) pathway and lignin degradation [50], another biosynthetic pathway would be responsible for the formation of the anisyl derivatives. Silk et al. also suggested these synthetic pathways of l-phenylalanine degradation by *Bjerkandera adusta*. They were also able to detect metabolites such as 3-phenyl pyruvic/phenylacetic/mandelic/benzoyl formic acid to support this hypothesis [50]. Such derivatives have not been detected for PSA. Therefore, further research is needed to get a full picture of the formation pathways in higher fungi.

## 3. Materials and Methods

### 3.1. Chemicals

Chemicals for the cultivation were supplied by AppliChem (Darmstadt, Germany), Alfa Aesar (Kandel, Germany), Carl Roth (Karlsruhe, Germany), and Sigma-Aldrich (Taufkirchen, Germany).

Diethyl ether (99.5%) was purchased from BCH Bruehl-Chemikalien Handel (Bruehl, Germany) and *n*-pentane (99%) from Th. Geyer (Renningen, Germany). These solvents were distilled before use (separately or as mixture P/D, 1:1.12, *v*/*v*).

5,6,7,7a-Tetrahydro-4,4,7a-trimethyl-2(4*H*)-benzofuranone (95%) was supplied by abcr (Karlsruhe, Germany) and linalool (97%), *p*-methylacetophenone (95%), 2-methyl-*n*-butanol (98%), *p*-methoxyphenylacetone (98%), and terpinen-4-ol (97%) were purchased from Acros Organics (Geel, Belgium). (2*E*)-Dec-2-enal (95%), 3,4-dimethoxybenzaldehyde (99%), hept-1-en-3-ol (98%), (*E*)-methyl cinnamate (99%), and *β*-pinene (99%) were obtained from Alfa Aesar (Kandel, Germany). Benzaldehyde (99%) was purchased from AppliChem (Darmstadt, Germany). Dimethyl sulfoxide (DMSO, 99.5%), ethanol (99.8%), isopropyl alcohol (99.8%), magnesium sulfate heptahydrate (MgSO_4_·7H_2_O), sabinene (98%), sodium chloride, sodium sulfate*,* and *α*-terpineol (96%) were purchased from Carl Roth and 1-(4-methoxyphenyl)-1,2-propanedione (95%) from ChemPUR (Karlsruhe, Germany). Acetaldehyde (99.5%), di-potassium hydrogen phosphate, and *α*-pinene (97%) were obtained from Fisher Scientific (Schwerte, Germany). *p*-Anisaldehyde (98%) and (*Z*)-linalool oxide (furanoid) (97%) were supplied by Fluka (Seelze, Germany) and Dess-Martion-Periodinane (97%) was acquired from Fluorochem (Hadfield, United Kingdom). *n*-Hexane (97%) was obtained from Honeywell (Offenbach, Germany) and (*R*)-limonene (100%) and l-tyrosine (98%) from Merck (Darmstadt, Germany). *m*-Anisaldehyde (97%), anisole (99%), chloroform-^2^H [99.8 atom%, with 0.03 vol% tetramethylsilane (TMS), stabilized with Ag], *p*-cymen-8-ol (95%), (*E*)-dihydrocarvone (98%), *n*-heptane, (*Z*)-limonene oxide (97%), methanol (99.9%), *p*-methoxybenzyl alcohole (98%), *p*-methoxypropiophenone (99%), 2-methylbut-3-en-2-ol (98%), piperitone (98%), propane-1,2-diol (99.5%), sodium pyruvate, sodium pyruvate-2-^13^C (99 atom% ^13^C), thiamine diphosphate (ThDP), and l-2-amino-3-([3,5-^2^H]-4-hydroxyphenyl)-propanoic acid (98 atom%) were purchased from Sigma Aldrich. *p*-Cymene (95%) and methyl-*p*-anisate (99%) were supplied by TCI (Zwijndrecht, Belgium). Acetonitrile (99.9%) and potassium dihydrogen phosphate were purchased from Th. Geyer and [7- ^13^C]-*p*-anisaldehyde (99.9 atom%; 97%) from Toronto Research Chemicals (Toronto, Canada). 3,6-Dimethyl-3a,4,5,7a-tetrahydro-1-benzofuran-2(3*H*)-one was kindly provided by Nils H. Schebb, University of Wuppertal.

Helium (5.0) as well as hydrogen (5.0) were purchased from Praxair (Düsseldorf, Germany) and nitrogen (5.0) from Air Liquide (Düsseldorf, Germany).

### 3.2. Citrus Side Stream

Two batches of *Citrus* side stream (CSS) were provided by the Spanish food processing industry, which were lyophilized (Martin Christ, Osterode am Harz, Germany) and milled (Vorwerk, Wuppertal, Germany) prior to use.

### 3.3. Fungal Strains

The dikaryotic strain (Dk) of PSA (DSM No. 8266) was obtained from Leibniz Institute DSMZ-German Collection of Microorganisms and Cell Cultures (Braunschweig, Germany). Thirty monokaryotic strains (Mk) derived from basidiospores of the Dk, a second Dk (Dk 3174), and 30 Mk derived from basidiospores of Dk 3174 (collection No. 343) were provided by the Institute of Food Chemistry, Leibniz University Hannover, Germany [19,63].

### 3.4. Cultivation of Pleurotus sapidus

PSA strains were kept as stock cultures on standard nutrition solution (SNS) agar plates according to Fraatz et al. or malt extract (ME) agar plates. SNS agar was composed of agar (15 g), l-asparagine monohydrate (4.5 g), potassium dihydrogen phosphate (1.5 g), magnesium sulfate hydrate (0.5 g), yeast extract (3.0 g), trace element solution [1 mL; copper(II) sulfate pentahydrate (5 mg L^−1^), iron(III) chloride hexahydrate (80 mg L^−1^), zinc(II) sulfate heptahydrate (90 mg L^−1^), manganese(II) sulfate hydrate (30 mg L^−1^), and ethylenediaminetetraacetic acid (0.4 g L^−1^)], and d-glucose monohydrate (30 g) (pH 6.0; respective compound per liter of deionized water) [64]. ME was composed of malt extract (20 g) and agar (15 g) (per liter of deionized water).

For the surface cultures, agar plates were prepared with *Citrus* side stream medium (CSSM) composed of agar (15 g L^−1^), potassium dihydrogen phosphate (1.5 g L^−1^), magnesium sulfate hydrate (0.5 g L^−1^), yeast extract (0.3 g L^−1^), trace element solution (1 mL L^−1^, same as described for SNS), CSS (amount depending on carbohydrate content, expressed as d-glucose monohydrate; target d-glucose monohydrate content in medium from CSS, 12 g L^−1^) (pH 6.0).

For submerged cultures, the following media were used (per liter of deionized water): SNS and CSSM as described above without agar; malt extract-peptone medium (MEP): malt extract (30 g) and peptone from soybean meal (3 g) (pH 5.6).

For preparation of precultures, a square with an edge length of 0.5 cm of an 80% overgrown agar plate was transferred to an Erlenmeyer flask (40% filling volume, *v*/*v*) followed by homogenizing at 10,000 rpm for 30 s using an Ultra Turrax T25 homogenizer (IKA, Staufen, Germany). The cultures were grown at 24 °C and 150 rpm in darkness on a rotary shaker (25 mm shaking diameter; Orbitron, Infors, Einsbach, Germany) for 6 days (SNS) and 4 days (MEP), respectively.

For the main culture, the respective medium was inoculated with 10% (*v*/*v*) homogenized preculture (30 s, 10,000 rpm) in Erlenmeyer flasks (40% filling volume, *v*/*v*). The cultures were grown under the same conditions as described for the precultures for up to 10 days.

### 3.5. Screening of Surface Cultures

For the surface cultures, a piece of mycelium of 80% overgrown SNS agar plate of both Dk and each Mk was cut out using a cork borer and was transferred to the CSSM agar plate. The cultures were incubated at 24 °C in darkness. From culture day 5 on or when the plate was at least 60% overgrown, the culture was subjected to sensory analysis. If the plate was more than 90% covered, it was evaluated for at least 5 more days.

### 3.6. Liquid/Liquid Extraction, Solvent-Assisted Flavor Evaporation, and Sensory Analysis

The precultures of PSA Dk and Mk 37, 74, 93, and 124 were grown in SNS medium, and the main cultures with CSSM in 250 mL Erlenmeyer flasks (40% filling volume, *v*/*v*) were produced in duplicates. On culture day 4, 10% sodium chloride (*w*/*v*) was added and the culture broth was stored at −4 °C until extraction. For aroma analysis, the samples were mixed with 50 µL each of internal standard *m*-anisaldehyde (0.74 mg mL^−1^ *n*-heptane and hept-1-en-3-ol (0.56 mg mL^−1^ *n*-heptane) and 40 mL distilled *n*-pentane/diethyl ether (P/D, 1:1.12, *v*/*v*). The liquid/liquid extraction was carried out by means of a magnetic stirrer (maximum stirring rate) for 30 min. The organic phase was separated from the aqueous phase via centrifugation (3500× *g*, 15 min, 4 °C). The extraction and separation steps were repeated twice. The organic phases were combined and the volatiles were separated by means of solvent-assisted flavor evaporation (SAFE) using a BAENG distillation unit (Bahr, Manching, Germany) connected to a high-vacuum pump (Pfeiffer Vacuum, Asslar, Germany). The purification was done under the following conditions: vacuum, 4.0 × 10^−4^ to 4.0 × 10^−3^ mbar; head and leg temperature, 55 °C; water bath temperature, 48 °C. Afterwards the purified organic extract was dried over anhydrous sodium sulfate and concentrated to a volume of about 1.5 mL using a Vigreux column (water bath temperature, 43 °C).

An aliquot (5 µL) of each of the SAFE distillate of the four selected Mk and the Dk was taken and the solvent was removed. The respective residue was dissolved in propane-1,2-diol (1.5 mL) and provided to the panelists (*n* = 16) in the form of a sniffing stick. Each sample was evaluated by the sensory panel for the intensity (0 = not perceptible to 5 = strong perceptible) of 10 predetermined odor attributes (citrus, buttery, anisic, vanilla, woodruff, woody, fruity, herbaceous, flowery, and sweetish). Finally, the median of the 16 ratings was calculated for each attribute of each culture extract. Additionally, the panelists were asked for their overall odor impression (‘---‘ = very unpleasant to ‘+++’ = very pleasant).

### 3.7. Aroma Extract Dilution Analysis

Aroma extract dilution analysis (AEDA) was carried out with the SAFE distillate of Mk 37 (cf. Section 3.6). It was diluted stepwise 1 + 1 (*v*/*v*) with P/D and analyzed using gas chromatography-flame ionization detection-olfactometry (GC-FID-O) by three panelists (1 male, 2 female) until no odorants were perceived at the olfactory detection port. If the odorant was perceived by all of the panelists, the FD factor was indicated as median. Otherwise, the mean value was calculated. If this was between two FD factors, the lowest value was given.

For the analysis, an Agilent (Agilent Technologies, Waldbronn, Germany) 7890A gas chromatograph equipped with an Agilent G4513A autosampler (sample volume 1 µL; cool-on-column injection) and an Agilent VF-WAXms column (30 m × 0.32 mm, 0.25 µm film thickness, temperature program, 40 °C, held for 3 min, ramped to 240 °C with 5 °C min^−1^, held for 12 min) was applied. After the column, the carrier gas hydrogen (2.2 mL min^−1^ constant flow) was split 1:1 (GERSTEL µFlow Manager) (GERSTEL, Mülheim an der Ruhr, Germany) to a flame ionization detector (temperature 250 °C; hydrogen 40 mL min^−1^; air 400 mL min^−1^; nitrogen 25 mL min^−1^) and a GERSTEL ODP 3 (transfer line 250 °C; mixing chamber 150 °C; make up gas nitrogen).

### 3.8. Identification and Structure Elucidation

For the identification of the flavor compounds, the LLE/SAFE distillate was analyzed additionally by means of gas chromatography-mass spectrometry using a polar or non-polar column. The MS and the retention indices (RI), calculated according to van den Dool and Kratz, of the substances were compared to those of authentic standards measured on the two different columns, published data, and the NIST 2011 MS library [65]. Furthermore, the perceived odor impressions were compared to those of published data and authentic standards.

The instrumental analysis was performed on the one hand with an Agilent 7890A gas chromatograph equipped with a MPS robotic multipurpose autosampler and coupled with an Agilent 7000B tandem mass spectrometer (GC-MS/MS). Prior to separation with a polar Agilent VF-WAXms column (30 m × 0.25 mm, 0.25 µm film thickness, temperature program, cf. Section 3.7) with a constant flow rate of 1.56 mL min^−1^ (carrier gas helium), the sample was introduced into the system via a split/splitless (S/SL) inlet (250 °C). The gas flow was split 1:1 using GERSTEL µFlow Manager into the mass spectrometer (ionization energy 70 eV; ion source 230 °C; quadrupoles 150 °C; transfer line 250 °C; scan in q1 *m/z* 33–300) and an olfactory detection port (ODP 3; transfer line 250 °C; mixing chamber 150 °C; make up gas nitrogen).

The gas chromatographic system used for the measurements with a non-polar column was an Agilent 7890A gas chromatograph connected to an Agilent 5975C mass spectrometer (GC-MS) (ionization energy 70 eV; ion source 230 °C; quadrupole 150 °C; transfer line 250 °C; scan mode *m/z* 33–300) equipped with an Agilent DB-5ms column (30 m × 0.25 mm, 0.25 µm film thickness, temperature program, 40 °C (3 min), heating to 300 °C (10 min) with 5 °C min^−1^; carrier gas helium; constant flow 1.2 mL min^−1^).

For identification of substance **48**, Mk 37 was cultivated in CSSM. The medium was extracted according to Section 3.6 but scaled up using Erlenmeyer flasks with 500 mL volume and harvested on day 8. After concentration using a Vigreux column, the extract was pre-fractionated by means of column chromatography (silica gel 60, Macherey-Nagel, Düren, Germany) with *n*-pentane and stepwise increased amounts of diethyl ether (98/2, 80/20, 50/50, 30/70, 0/100, *v*/*v*) as eluents. The fractions were concentrated using a Vigreux column to about 1.5 mL, and the fraction containing the target substance was purified by means of preparative HPLC using an YL9110S HPLC system (Young Lin, Anyang, South Korea). The separation was carried out on a Nucleodur 100-5 column, 250 × 21 mm^2^, with a corresponding guard column (10 × 16 mm^2^) (Macherey-Nagel), using *n*-hexane/isopropyl alcohol, 99/1, *v/v* in isocratic mode (15 mL min^−1^). Substances were detected using an YL9120S ultraviolet/visible (UV/VIS) detector at 230 nm and 265 nm and collected via a CHF 122SC fraction collector (Advantec, Dublin, OH, USA; fraction volume 5 mL). Structure elucidation was performed by means of GC-MS, electrospray ionization-time of flight-mass spectrometry (ESI-TOF-MS), and nuclear magnetic resonance (NMR) spectroscopy.

The accurate mass was determined using an AB Sciex (Darmstadt, Germany) TripleTOF 5600+ system with direct infusion (10 µL min^−1^) of the isolated product dissolved in methanol. Ionization was performed in positive mode (nebulizer gas 15 psi; auxiliary gas 15 psi; curtain gas 25 psi; source temperature 0 °C; ion spray voltage floating 5500 V). The *m/z* scan range of the mass spectrometer, which was calibrated using sodium formate solution, was set between 100 and 2000 Da. Data were analyzed using Analyst TF 1.7.1 and PeakView 2.2 software (AB Sciex).

NMR experiments [^1^H NMR,^13^C NMR, DEPT 135, ^1^H, ^1^H correlation spectroscopy (COSY), heteronuclear single-quantum correlation (HSQC), heteronuclear multi-bond correlation (HMBC)] were carried out on Bruker Avance II 400 MHz and Bruker Avance III 400 MHz HD spectrometers (Rheinstetten, Germany), using CDCl_3_ as solvent.

Chiral HPLC was performed using an HPLC system (Knauer, Berlin, Germany) fitted with a diode array detector (DAD 2.1L) and equipped with a chiral phase column Chiralpak IC (250 mm × 4.6 mm i.d., *n*-hexane/ethylacetate/formic acid, 85/15/0.05, *v*/*v*/*v*; flow 1 mL min^−^^1^; room temperature; 270 nm)

In addition, 2-HPP was prepared according to Nakamura et al. via reduction of 1-(4-methoxyphenyl)-1,2-propanedione with baker’s yeast [55]. Therefore, 2 g of commercially available baker’s yeast was mixed with 20 mL phosphate buffer (50 mM, pH 7.0, DMSO, 20%, thiamine diphosphate (ThDP), 2 mM, and magnesium sulfate heptahydrate (MgSO_4_·7H_2_O), 20 mM) and pre-incubated at 24 °C, 150 rpm, for 65 min. Afterwards, an aliquot (1.5 mL) of the yeast suspension was added to 5 mg of the diketone. The incubation was performed in glass vials (4 mL) on a rotary shaker (40 rpm, IKA) at 30 °C for 150 min. Subsequently, the samples were extracted after adding 2 mL P/D on a rotary shaker at 40 rpm for 10 min, followed by centrifugation (3500× *g*, 10 min, 4 °C). The organic phase was dried over anhydrous sodium sulfate and analyzed by means of gas chromatography-mass spectrometry.

The absolute configuration of the produced standard was determined by measuring the sample (1.6 mg in 1 mL acetonitrile) by means of a polarimeter P-2000 (JASCO, Pfungstadt, Germany) at 589 nm at room temperature.

### 3.9. Stable Isotope Dilution Analysis of p-Anisaldehyde

Cultures for the stable isotope dilution analysis (SIDA) were prepared in the same way as for the AEDA (Mk 37, 100 mL culture volume + 10% inoculum, *v*/*v*). Instead of CSS, l-tyrosine (33 mM) was used as substrate in a medium containing 15 g L^−1^ d-glucose.

Cultures were prepared as triplicates. On day 4, 6, and 8, an aliquot of 2 mL was taken under sterile conditions. Depending on the expected content of *p*-anisaldehyde, different amounts of [7-^13^C]-*p*-anisaldehyde stock solution were added (Table 3).

Subsequently, 2 mL P/D was added and the mixture was homogenized (Vortex, Barneveld, WI, USA, 30 s), followed by extraction (10 min, 40 rpm, 24 °C) and centrifugation (3500× *g*, 10 min, 4 °C). This step was repeated once. The organic phases were pooled and dried over anhydrous sodium sulfate. An aliquot of 500 µL of the LLE was supplemented with 50 µL of internal standard solution (*m*-anisaldehyde, 1 mg mL^−1^) prior to analysis by means of GC-MS.

For this purpose, an Agilent 7890B gas chromatograph equipped with a MPS robotic multipurpose autosampler was used. The sample was introduced into the system via a split/splitless (S/SL) inlet (250 °C). The same polar VF-WAXms column and temperature program as described above were employed with a constant flow rate of 1.2 mL min^−1^ (helium). The compounds were detected using a mass spectrometer (Agilent 5977B, ionization energy 70 eV; ion source 230 °C; quadrupole 150 °C; transfer line 250 °C; scan mode *m/z* 33–300).

### 3.10. Supplementation of l-Tyrosine

Tyrosine supplementation studies were carried out in 100 mL Erlenmeyer flasks (40 mL filling volume + 10% SNS inoculum) according to Section 3.4, but using l-2-amino-3-([3,5-^2^H]-4-hydroxyphenyl)-propanoic acid (33 mM) instead of CSS, and the amount of glucose was increased to 15 g L^−1^. The cultures (Dk, Mk 37) were harvested after 6 days, extracted by means of LLE (cf. Section 3.6), and measured via GC-MS (cf. Section 3.9)

### 3.11. Synthesis of 1-(4-Methoxyphenyl)-1,[2-^13^C]-Propanedione and [2-^13^C]-HPP

The synthesis of 1-(4-methoxyphenyl)-1,[2-^13^C]-propanedione was carried out in two steps starting from *p*-anisaldehyde (50 mM) and ^13^C_2_ sodium pyruvate (25 mM), which were transformed using lyophilized PSA mycelium (Dk, DSM No. 8266) to the α-hydroxy ketones. Therefore, PSA was cultivated in MEP medium (ME stock culture, MEP preculture) and harvested after 3 days. The mycelium was separated from the supernatant via vacuum filtration using a Büchner funnel and washed with sodium chloride solution (0.9%, *w*/*v*). Afterwards, the mycelium was freeze-dried (Martin Christ) and stored at −80 °C until usage.

The reaction mixture of 20 parallel samples was composed of 50 mg lyophilisate, 50 mM phosphate buffer (pH 7.0), substrates, dimethyl sulfoxide (DMSO, 20%), thiamine diphosphate (ThDP, 2 mM), and magnesium sulfate heptahydrate (MgSO_4_·7H_2_O, 20 mM); the final volume was 1.5 mL. The incubation was carried out in glass vials (4 mL) on a rotary shaker at 40 rpm (24 h, 30 °C) [66,67]. After incubation, the samples were extracted by adding 2 mL P/D (1:1.12, *v*/*v*) on a rotary shaker at 40 rpm for 10 min, followed by centrifugation (3500× *g*, 10 min, 4 °C). The organic phase was dried over anhydrous sodium sulfate and analyzed by means of gas chromatography-mass spectrometry (cf. Section 3.9). After removal of the solvent using a Vigreux column, 75 mL dichloromethane and 1.8 g Dess-Martin-Periodinane were added to the product and stirred for approx. 2 h at room temperature [68,69]. Subsequently, the diketone was separated stepwise. For this, 30 mL of the reaction mixture was extracted with 60 mL 1 M sodium hydroxide solution and 150 mL P/D (1:1.12, *v*/*v*) with a separating funnel. After separation and washing, the organic phases were combined, dried over anhydrous sodium sulfate, and filtered. Subsequently, the solvent was removed by means of a Vigreux column and the product was purified using a preparative AZURA HPLC system (Knauer, Berlin, Germany). The separation was carried out on a Nucleosil 100-7 C18 column, 250 mm × 16 mm i.d. with corresponding guard column (10 mm × 16 mm i.d.) (Macherey Nagel) using acetonitrile/water (1/1, *v*/*v*) in isocratic mode (8 mL min^−1^, P 6.1L). Substances were detected using a UVD 2.1S ultraviolet/visible (UV/VIS) detector at 294 nm and fractions were collected using a Foxy R1 fraction collector (fraction volume, 5 mL). The purity was evaluated by means of GC-MS/MS (cf. Section 3.8). The reduction of the substituted diketone [to [2-^13^C]-HPP by lyophilized PSA Mk 37 mycelium (MEP)] was carried out analogously to the procedure of the reaction with baker’s yeast described in Section 3.8.

### 3.12. Approximation of the Odor Threshold of 2-HPP

Approximation of the odor threshold of 2-HPP (OT_2-HPP_) was performed according to Ullrich and Grosch using (2*E*)-dec-2-enal with an odor threshold in air of 2.7 ng L^−1^ as internal standard [Equation (1)] [70,71].
(1)$${\mathrm{OT}}_{2-\mathrm{HPP}}=\frac{{\mathrm{OT}}_{I}•C_{2-\mathrm{HPP}}•D_{I}}{C_{I}•D_{2-\mathrm{HPP}}}$$

OT_2-HPP_: odor threshold of 2-HPP; OT_I_: odor threshold of internal standard; C_I_: concentration of internal standard; C_2-HPP_: concentration of 2-HPP; D_I_: D-value of internal standard; D_2-HPP_: D-value of 2-HPP. The D-value represents the highest dilution at which a substance is still smelled [70].

A mixture of isolated 2-HPP (cf. Section 3.8) and the internal standard (each 0.5 mg L^−1^) was stepwise diluted 1 + 1 (*v*/*v*) with acetonitrile and analyzed by means of GC-FID-O (cf. Section 3.7; deviating temperature program, 40 °C (3 min), heated to 200 °C (8 min) with 5 °C min^−1^, heated to 240 °C (7 min) with 20 °C min^−1^; deviating temperatures ODP, transfer line, 300 °C; mixing chamber, 200 °C). The analysis was carried out by three panelists (1 female, 2 male). The proper dilution of the stock solution was controlled by plotting the log_2_ of the peak areas of the internal standard and the target substance against the dilution factor. 1-HPP and the diketone as GC-related artefacts were also considered (Appendix A).

## 4. Conclusions

Out of the 60 Mk of PSA grown on *Citrus* side streams, Mk 37 was identified as a promising candidate, which produced a particularly intense woodruff and anise-like overall aroma. AEDA of SAFE/LLE of a liquid culture showed that, apart from other arylic substances, mainly *p*-anisaldehyde shaped the pleasant odor impression and was, as revealed via SIDA, present in high concentrations of up to 160 mg L^−1^ on the 8th culture day.

Apart from that, the acyloin 2-HPP contributed to the overall aroma of the cultures. This compound was isolated and structurally characterized. The odor of 2-HPP (herbaceous and sweetish) was described for the first time in this study, and a comparatively low odor threshold of 0.2 ng L^−1^ to 2.4 ng L^−1^ (air) was determined for the almost enantiopure α-hydroxyketone. Its biosynthesis, like that of anisaldehyde, was traced back to the precursor l-tyrosine by means of isotopically substituted l-tyrosine. For the first time, a reduction of the corresponding diketone via PSA was also shown.

## Figures and Tables

**Figure 1 molecules-27-00651-f001:**
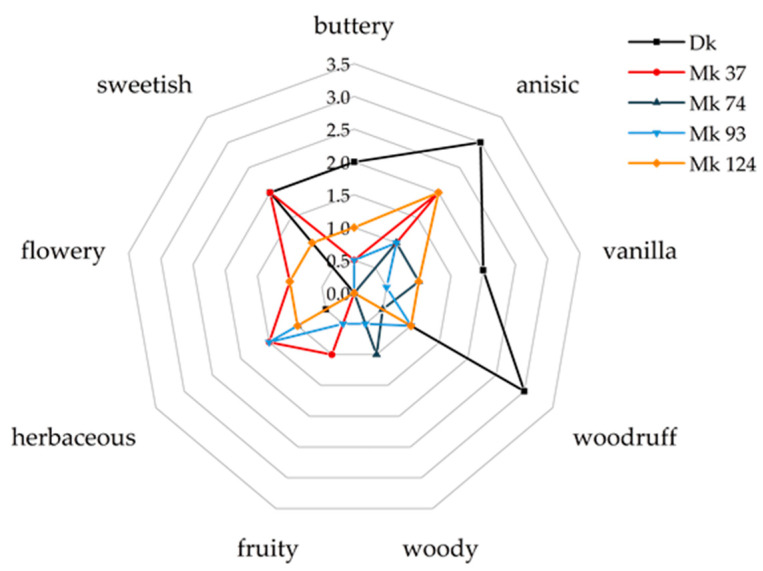
Olfactory impressions of LLE/SAFE samples from CSSM cultures of a dikaryotic strain (Dk) and different monokaryotic strains (Mk).

**Figure 2 molecules-27-00651-f002:**
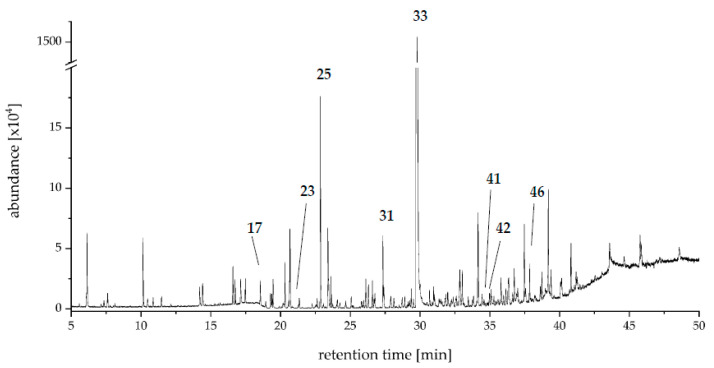
GC-MS chromatogram (VF-WAXms) from LLE/SAFE CSSM PSA Mk 37 culture harvested on culture day 4.

**Figure 3 molecules-27-00651-f003:**
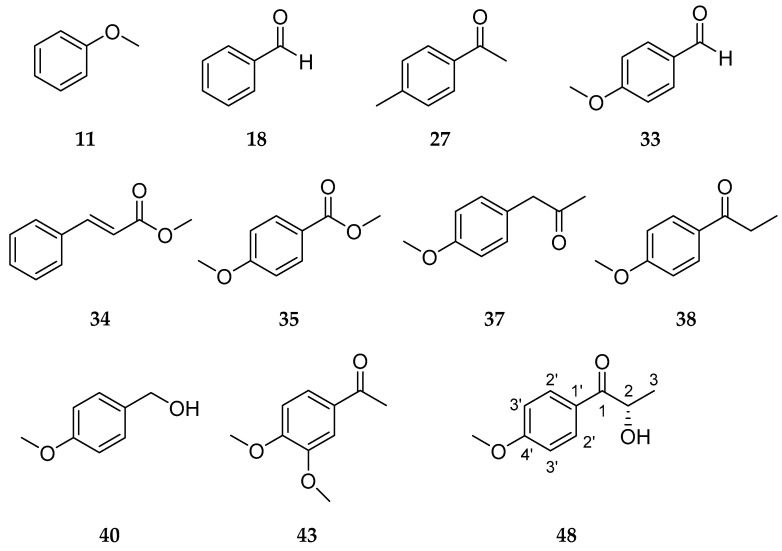
Structures of arylic compounds identified in CSSM PSA Mk 37.

**Figure 4 molecules-27-00651-f004:**
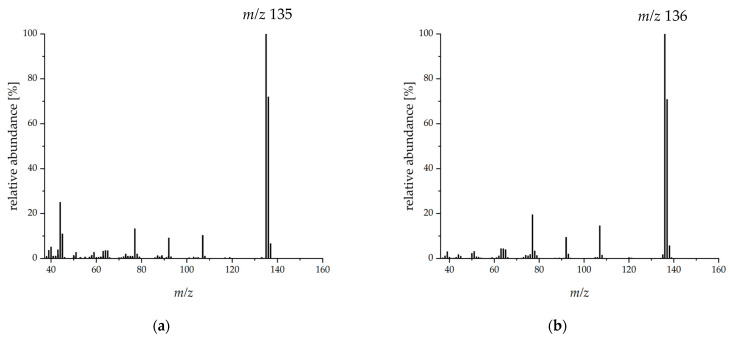
Mass spectra of *p*-anisaldehyde (**a**) and of [7-^13^C]-*p*-anisaldehyde (**b**).

**Table 1 molecules-27-00651-t001:** Results of aroma extract dilution analysis of CSSM PSA Mk 37 by means of LLE/SAFE.

No.	RI		Compound	Perceived Odor	FD Factor	Identification ^1^
	VF-WAXms	DB-5 ms				
**1**	1016	930	*α*-pinene	herbaceous, sweetish, terpenic	128	MS, RI, STD, O
**2**	1036	<800	2-methylbut-3-en-2-ol	herbaceaous	2048	MS, RI, STD, O
**3**	1062		Ni ^4^	sweetish, fruity	128	
**4**	1102	974	*β*-pinene	herbaceous, green, sourish	512	MS, RI, STD, O
**5**	1120	970	sabinene	terpenic, herbaceous, musty	2048	MS, RI, STD, O
**6**	1197	1027	(*R*)-limonene	citrus, fresh	256	MS, RI, STD, O
**7**	1209		2-methyl-*n*-butanol	green, sweetish	512	MS, RI, STD, O
**8**	1268	1023	*p*-cymene	citrus, fresh, terpenic	8192	MS, RI, STD, O
**9 ^2^**	1296		ni	green, herbaceous	512	
**10 ^2^**	1340		ni	minty, sweetish	512	
**11**	1345	913	anisole	flowery, sweetish	1024	MS, RI, STD, O
**12**	1355	878	hept-1-en-3-ol (IS)	green, spicy	256	
**13 ^2,3^**	1375		ni	citrus, herbaceous	16,384	
**14 ^2,3^**	1424		ni	musty, terpenic	256	
**15 ^3^**	1448	1131	(*Z*)-limonene oxide	flowery, sweetish	1024	MS, RI, STD, O
**16**	1469	1085	(*Z*)-linalool oxide (furanoid)	citrus, herbaceous, sweetish	64	MS, RI, STD, O
**17**	1510	1184	3,6-dimethyl-2,3,3a,4,5,7a-hexahydrobenzofuran (dill ether)-isomer	green, citrus, sweetish	32,768	MS, RI, STD ^5^, O
**18**	1530	958	benzaldehyde	sweetish	1024	MS, RI, STD, O
**19**	1550	1098	linalool	flowery, fruity	2048	MS, RI, STD, O
**20**	1593	1229	3,6-dimethyl-2,3,3a,4,5,7a-hexahydrobenzofuran (dill ether)-isomer	minty, citrus	128	MS, RI, STD ^5^, O
**21**	1601	1179	terpinen-4-ol	terpenic, green	1024	MS, RI, STD, O
**22 ^3^**	1608	1196	dihydrocarvone			
**23**	1630		ni	flowery, fresh	32,768	
**24**	1634	1148	ni	herbaceous	8192	
**25**	1697	1193	*α*-terpineol	green, herbaceous, flowery	16,384	MS, RI, STD, O
**26**	1729	1255	piperitone	citrus, flowery, minty	64	MS, RI, STD, O
**27 ^3^**	1763	1182	*p*-methylacetophenone	sweetish	2048	MS, RI, STD, O
**28**	1782		ni	green, fatty	2048	
**29**	1850	1185	*p*-cymen-8-ol	sweetish, flowery, citrus	512	MS, RI, STD, O
**30**	1881		ni	flowery, sweetish, green	1024	
**31**	1908	1196	*m*-anisaldehyde (IS)	flowery, sweetish	512	
**32**	1988		ni	green, sweetish, woodruff	512	
**33**	2034	1265	*p*-anisaldehyde	anisic, sweetish, woodruff	262,144	MS, RI, STD, O
**34**	2080	1382	(*E*)-methyl cinnamate	balsamic, sweetish	1024	MS, RI, STD, O
**35**	2095	1371	methyl-*p*-anisate	sweetish, herbaceous	4096	MS, RI, STD, O
**36**	2121		ni	minty, sweetish, vanilla	512	
**37**	2155	1379	*p*-methoxyphenylacetone	green, woody, spicy	512	MS, RI, STD, O
**38**	2211	1447	*p*-methoxypropiophenone	sweetish, fruity, green	1024	MS, RI, STD, O
**39**	2240	1449	3,6-dimethyl-3a,4,5,7a-tetrahydro-1-benzofuran-2(3*H*)-one (wine lactone)-isomer	sweetish, fruity, green	2048	MS, RI, STD, O
**40**	2276	1281	*p*-methoxybenzyl alcohol	sweetish, herbaceous	512	MS, RI, STD, O
**41**	2298		ni	flowery, waxy, sweetish	16,384	
**42**	2350	1524	ni	flowery, sweetish, herbaceous	65,536	
**43**	2400	1474	3,4-dimethoxybenzaldehyde	sweetish, woodruff	8192	MS, RI, STD, O
**44**	2427		ni	sweetish, fruity	256	
**45**	2461		ni	flowery, vanilla, herbaceous	128	
**46**	2508		ni	herbaceous, spicy, flowery	16,384	
**47**	2540		ni	fruity, herbaceous, sweetish	1024	
**48**	2571	1549	2-HPP	herbaceous, sweetish	512	MS, NMR, HRMS

^1^ Identification via comparison of measured mass spectra with NIST 2011 database (MS), comparison of the calculated retention indices with published data on a polar and a non-polar column (RI), confirmation of mass spectra and retention indices on both columns with authentic standards (STD), comparison of perceived odor impression with literature and authentic standard (O) and by means of NMR and HRMS after isolation of the respective compound; ^2^ RI calculated for the perceived odor; ^3^ odor intensities might have been synergistically enhanced by an coeluting compound; ^4^ not identified (ni); ^5^ for the identification, an extract of linden blossom honey was used.

**Table 2 molecules-27-00651-t002:** Concentrations of *p*-anisaldehyde in CSS Medium PSA Mk 37 depending on the culture day.

Substrate	Culture Day 4	Culture Day 6	Culture Day 8
	Concentration [mg L^−1^]		
CSS ^1^	9.2 ± 1.7	147.0 ± 31.8	160.3 ± 26.9

^1^ CSS: *Citrus* side stream.

**Table 3 molecules-27-00651-t003:** Amounts of [7-^13^C]-*p*-anisaldehyde stock solution added to culture aliquots prior to extraction.

Substrate	Culture Day 4			Culture Day 6			Culture Day 8		
	I	II	III	I	II	III	I	II	III
	Volume [µL]								
CSS ^1^	10	10	10	200	200	100	300	150	200
l-tyrosine	20	20	30	80	120	100	100	150	80

^1^ CSS: *Citrus* side stream.

## Data Availability

The data presented in this study are available on request from the corresponding author. The data are not publicity available due to the large data set.

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
