# Peer review of "Production of an Anise- and Woodruff-like Aroma by Monokaryotic Strains of Pleurotus sapidus Grown on Citrus Side Streams"

_molecules, 2022, doi:10.3390/molecules27030651_

Round 1

Reviewer 1 Report

This manuscript is well constructed and thoroughly supported by experimental data, within the limits of its claims. The novelty is moderate as rather than presenting structure elucidation of newly identified compounds, it only presents the identification of a known compound with a new (olfactory/aroma) attribute. Some insight, supported by experiments, into the biosynthesis of this and related compounds is also provided.

Overall, the manuscript is well presented and the English level is high; I believe, however, that a short conclusion is necessary, summarizing in one paragraph all essential findings and how they are supported by the experiments.

Some minor aspects:

-Table S1 is missing - at least from the supplementary files zip archive I downloaded from the journal platform; please make sure to include this supporting table in the supplemental part

-Lines 118-119: "Figure 2" should read "Figure 1" here

Author Response

- I believe, however, that a short conclusion is necessary, summarizing in one paragraph all essential findings and how they are supported by the experiments.

As suggested by the reviewer, a short conclusion was added as paragraph 5:

“5. Conclusions

Out of the 60 Mk of PSA grown on Citrus side streams, Mk 37 was identified as a promising candidate, which produced a particularly intense woodruff and anise-like overall aroma. AEDA of SAFE/LLE of a liquid culture showed that, apart from other arylic substances, mainly p-anisaldehyde shaped the pleasant odor impression and was, as revealed by SIDA, present in high concentrations of up to 160 mg L-1 on the 8th culture day.

Apart from that, the acyloin 2-HPP contributed to the overall aroma of the cultures. This compound was isolated and structurally characterised. The odor of 2-HPP (herbaceous and sweetish) was described for the first time in this study, and a comparatively low odor threshold of 0.2 ng L-1 to 2.4 ng L-1 (air) was determined for the almost enantiopure α-hydroxyketone. Its biosynthesis, like that of anisaldehyde, was traced back to the precursor l-tyrosine by means of isotopically substituted l-tyrosine. For the first time, a reduction of the corresponding diketone by PSA was also shown.

-Table S1 is missing - at least from the supplementary files zip archive I downloaded from the journal platform; please make sure to include this supporting table in the supplemental part

            We are sorry for that. Table S1 has now been included in the zip archive.

-Lines 118-119: "Figure 2" should read "Figure 1" here

            Section 2.2 reports on the individual substances shaping the overall flavor. Figure 2 shows the GC-MS chromatogram of the analyzed culture extract; the reference to Figure 2 was now inserted after the second sentence of this section for clarification (lines 117 and 118).

Reviewer 2 Report

The manuscript by Burger et al. describes the analysis of aromas produced by mono and dikaryotic strains of Pleurotus sapidus. This kind of comparative analysis is not very common and, therefore, they have scientific interest. The experiments were carried out initially in solid culture medium, under controlled conditions. In experiments with solid medium (CSSM plus agar), sensory analysis was conducted after 5 days of cultivation or after profuse fungal growth. After selecting the target strains, the experiments of cultivation were conducted in liquid medium (SNS) were processed by liquid-liquid extraction under magnetic agitation. This change in the culturing step was inserted to adapt the system to possible industrial applications, and it was a plus in the methodology. The volatile compounds present in the organic phases were separated by solvent assisted flavor evaporation, purified and subjected to panelists evaluation. The identification of the flavor compounds was accomplished using GC-MS and GC-MS/M. The results were very interesting; for example, p-anisaldehyde aroma was detected in very diluted extracts. The work was elegantly finished, with supplementation of the culture medium with an isotopically marked biosynthetic precursor for biosynthesis elucidation, and care was taken to determine the enantiomeric excess using HPLC with chiral column.

In general, the work was very well structured, the theme and results are interesting and the conclusions are supported by the results.

Some minor points/corrections are pointed below.

Line 59. Remove the extra full stop.

Line 60. I suggest replacing the colloquial term “terpenoid structures” by “compounds with terpenoid structures”

Results & Discussion

The results point to some strains producing anise, marzipan and vanilla-like aromas, while others were reported to produce woodruff-like, tonka bean, herbaceous or even unpleasant and musty scents. Some strains were selected, cultured in liquid medium and analyzed to verify quali- and quantitative differences between the strains.

Line 88. Please, make sure table S1 has been included in the supplemental material. I couldn't find it.

Line 145. Use “Compound 22” instead of 22; the same in line 207; please, verify the correctness of sentences starting with cardinal numbers.

Line 239. Please, abbreviate P. pulmonaris, since the genus has already been written in line 203. The same for P. ostreatus (line 63 x line 249)

Materials and Methods

Please, refer to Molecules Guide to the authors to verify the use of & (used in line 78 (“Results & Discussion”) or and, used in line 336 (“Materials and Methods”)

Line 386 (cultivation parameters). SNS and malt agar were used for PSA storage. Ok. As I understand, a screening in CSSM-agar (surface culture) was the initial experiment (line 80). Then (line 89), submerged cultures were set up. But in the Materials and Methods (from line 396), you describe the media composition for the submerged culturing before describing the screening using the surface cultures/media composition (described only in the item 4.5; line 414). Since the surface culturing was conducted prior to the submerged experiment, I suggest to write this part of the experimental section in a chronological order. Or, at least, transfer the composition of the media used for surface culturing to item 4.4. Just after line 395.

Line 396 again. Did you use three culture media? Agar-free SNS, Supplemented CSSM and MEP? If it is correct, I suggest to rephrase or, at least, substitute the comma by a colon, in lines 397 [after (CSSM)], and 402 [after (MEP)]. The commas confused the description.

Lines 397 and followings. I found it difficult to understand the rational for the culture media used. Why did you change the composition of the ME culture medium in the cultivation step? I mean, why peptone was added?

In general, why describing different media in the M&M section, since you presented the results only for CSSM? (Please, take into consideration that I did not find Table S1; these results may be there). I would suggest to revise carefully his part of the M&M section.

Line 412: Please, correct “ten 10”

Lines 511 and 513. Here you used the brackets to avoid double brackets. I suggest you do the same for other situations in the manuscript. For instance, in lines 190, 191, 193, and 393, instead of using “))”.

Line 442. How many panelists? 16 or 3, as in line 451? If they were three, wasn’t a small number of panelists for an accurate result? And, the procedure for panelists analysis were not described or referenced. Conditions reported in lines 445 and 446 are not enough to repeat the experiment.

Supplementary Material

Please, refer to Molecules Guide to the authors to verify the necessity (or not) to add legends for the spectra and graph.

Author Response

Line 59. Remove the extra full stop

            The extra full stop was removed, as suggested by the reviewer.

Line 60. I suggest replacing the colloquial term “terpenoid structures” by “compounds with terpenoid structures”

            As suggested by the reviewer, the term “terpenoid structures” was changed to “compounds with terpenoid structures”.

Results & Discussion

Line 88. Please, make sure table S1 has been included in the supplemental material. I couldn't find it.

            Thank you, Table S1 has been included in the zip archive.  

Line 145. Use “Compound 22” instead of 22; the same in line 207; please, verify the correctness of sentences starting with cardinal numbers.

            We verified all sentences starting with cardinal numbers and used “compound XX” instead of “XX”.

Line 239. Please, abbreviate P. pulmonaris, since the genus has already been written in line 203. The same for P. ostreatus (line 63 x line 249)

            As requested, we have now used the abbreviation for Pleurotus after the first mention of the genus (line 240 P. pulmonarius, line 250 P. ostreatus) in the revised document.

Materials and Methods

Please, refer to Molecules Guide to the authors to verify the use of & (used in line 78 (“Results & Discussion”) or and, used in line 336 (“Materials and Methods”)

               The headline “Results & Discussion” was changed to “Results and Discussion”.

Line 386 (cultivation parameters). SNS and malt agar were used for PSA storage. Ok. As I understand, a screening in CSSM-agar (surface culture) was the initial experiment (line 80). Then (line 89), submerged cultures were set up. But in the Materials and Methods (from line 396), you describe the media composition for the submerged culturing before describing the screening using the surface cultures/media composition (described only in the item 4.5; line 414). Since the surface culturing was conducted prior to the submerged experiment, I suggest to write this part of the experimental section in a chronological order. Or, at least, transfer the composition of the media used for surface culturing to item 4.4. Just after line 395.

As suggested, we have transferred the composition of the surface culture medium to section 4.4 (lines 397-402).

Line 396 again. Did you use three culture media? Agar-free SNS, Supplemented CSSM and MEP? If it is correct, I suggest to rephrase or, at least, substitute the comma by a colon, in lines 397 [after (CSSM)], and 402 [after (MEP)]. The commas confused the description.

For the submerged cultures we used the three media: agar-free SNS, CSSM and MEP. As recommended, the comma was substituted by a colon (line 409).

Lines 397 and followings. I found it difficult to understand the rational for the culture media used. Why did you change the composition of the ME culture medium in the cultivation step? I mean, why peptone was added?

In general, why describing different media in the M&M section, since you presented the results only for CSSM? (Please, take into consideration that I did not find Table S1; these results may be there). I would suggest to revise carefully his part of the M&M section.

               The standard protocol for submerged cultivation of basidiomycetes comprises 3 steps. It typically starts with an agar plate, on which the fungi are also stored. In the next stage, a so-called preculture is created, from which, however, no analyses are usually carried out. From this preculture, an inoculum is taken for the next step, the main culture. The desired analyses are then carried out from this.

In our case, two different submerged cultures were created. One was taken from the SNS agar plate via the SNS preculture to the CSSM main culture. From this culture, the AEDA analysis and quantification of anisaldehyde were performed.

The other cultivation started with the ME agar plate and was led via the MEP preculture to the MEP main culture. Lyophilized mycelia from these cultures were used for the biotransformation experiments. Peptone is often added to the malt extract in liquid cultivation as an additional and alternative nitrogen source for better growth of the fungi.

Line 412: Please, correct “ten 10”

The sentence was corrected and “ten” was deleted (line 419).

Lines 511 and 513. Here you used the brackets to avoid double brackets. I suggest you do the same for other situations in the manuscript. For instance, in lines 190, 191, 193, and 393, instead of using “))”.

We checked the manuscript for double brackets and used square brackets in all cases to avoid double brackets, as suggested by the reviewer.

Line 442. How many panelists? 16 or 3, as in line 451? If they were three, wasn’t a small number of panelists for an accurate result? And, the procedure for panelists analysis were not described or referenced. Conditions reported in lines 445 and 446 are not enough to repeat the experiment.

            There were two sensory panels in this context. In the first panel, the test persons used sniffing sticks to assess the overall aroma of the SAFE extracts of Dk, Mk 37, Mk 74, Mk 93 and Mk 124 (section 2.1 and 4.6). In this experiment, the 16 panelists were asked to rate the intensity of 10 predetermined odor attributes of the culture extract.  

In the following experiment, AEDA was carried out by 3 panelists with the distillate of Mk 37 (section 2.2 and 4.7), as this was the most interesting monokaryotic strain.

Supplementary Material

Please, refer to Molecules Guide to the authors to verify the necessity (or not) to add legends for the spectra and graph

            As far as we could see, there are no specific requirements for captions in the author guidelines. According to these guidelines, captions and legends should be clear, accurate and contain all relevant information. We have taken these requirements into account in all figures.

We appreciate the valuable reviewing process and hope to have answered all of the requests satisfactorily.

                   For the authors,

                                                                                    Holger Zorn